# Study on Corrosion Resistance and Conductivity of TiMoN Coatings with Different Mo Contents under Simulated PEMFC Cathode Environment

**DOI:** 10.3390/ma15113766

**Published:** 2022-05-25

**Authors:** Jie Jin, Rui Cao, Jinzhou Zhang, Yi Tao, Xu Tian, Xianli Kou

**Affiliations:** College of Materials Science and Engineering, Zhejiang University of Technology, Hangzhou 310000, China; jinjie@zjut.edu.cn (J.J.); dr_zhangjinzhou@126.com (J.Z.); taoy0526@163.com (Y.T.); txdawn@163.com (X.T.); kouxl2020@163.com (X.K.)

**Keywords:** TiMoN coating, bipolar plates, corrosion resistance, interfacial contact resistance, PEMFC

## Abstract

TiMoN coatings with different Mo contents on a SS316L substrate are deposited by using closed field unbalanced magnetron sputtering ion plating (CFUMSIP) technology to enhance the corrosion resistance and durability of stainless steel (SS) bipolar plates (BPs) in proton exchange membrane fuel cell (PEMFC) during the start-up/shut-down process. The electrochemical test results illustrate that TiMoN-4A coating has extremely good corrosion resistance compared to other coatings. The potentiostat polarization (+0.6 V_SCE_) tests indicate that the corrosion current density (I_corr_) of TiMoN-4A coating is 5.22 × 10^−7^A cm^−2^, which meets the department of energy 2020 targets (DOE, ≤1 × 10^−6^ A cm^−2^). Otherwise, TiMoN-4A coating also exhibits the best corrosion resistance and stability in potentiostatic polarization, electrochemical impedance spectroscopy (EIS), and high potential (+1.2V_SCE_) polarization tests. The interfacial contact resistance (ICR) measurement results show that TiMoN-4A coating has the minimum ICR of 9.19 mΩ·cm^2^, which meets the DOE 2020 targets (≤10 mΩ·cm^2^).

## 1. Introduction

Proton exchange membrane fuel cells (PEMFC) have the advantages of high conversion efficiency, environmentally friendly, low operating temperature, fast startup speed, high power density, high reliability, and small size [1,2,3,4,5]. In recent decades, PEMFC has attracted extensive attention in the fields of automobiles, airplanes, fixed power supplies, and mobile applications [6,7]. However, its mass production application is limited due to the high cost and poor durability under working conditions. As an important part of PEMFC, bipolar plates (BPs) account for 80–85% of the total mass and 25–45% of the cost [8,9]. The main functions of BPs are to collect and transmit electrons, connect a single membrane electrode assembly (MEA) series [10,11,12,13], connect the battery structure and power backbone, and channels for air, fuel, and coolant. In recent decades, stainless steel BPs have attracted widespread attention because of their advantages of high conductivity, good mechanical properties, strong airtightness, and low cost [14]. Unfortunately, under the influence of irregular voltage changes, harsh environment, and long-term immersion in the electrolyte, stainless steel suffers severe corrosion and passivation, which considerably increases the interfacial contact resistance (ICR) between BPs and the gas diffusion layer (GDL). In addition, metal corrosion can produce metal ions, which will damage the MEA and reduce the efficiency of PEMFC [15]. To withstand the corrosive environment of PEMFC, it is essential to investigate stainless steel BPs with great corrosion resistance and high conductivity.

Surface treatment of BPs is an efficient method to enhance corrosion resistance and conductivity [16]. Metal nitrides are usually manufactured by physical vapor deposition (PVD), which have the advantages of high hardness, high thermal stability, low friction coefficient, good wear resistance and excellent corrosion resistance [17], such as CrN [18], ZrN [19,20], Mo_2_N [21], TiN [22], TiSiN [23], TiAlN [24,25], CrAlN [26], etc. Nowadays, many kinds of transition metal nitride coatings have been widely studied and applied in PEMFC. In our previous research, He et al. [27] successfully prepared the TiN on Ti-6Al-4V alloy by using liquid-phase plasma electrolytic nitriding technology. The electrostatic potential polarization experiment demonstrated that the potential of the simulated PEMFC cathode environment (0.6 V_SCE_) has the lowest corrosion current density (5.7 × 10^−7^ A·cm^−2^). The interfacial contact resistance test indicated that its ICR was 6 ± 0.4 mΩ·cm^2^, which fully met the department of energy 2020 targets (DOE, ≤10 mΩ·cm^2^). Yan et al. [28] deposited TiN-Ag coatings on SS316L and aluminum at different temperatures by closed-field magnetron sputtering ion plating system. The consequences showed that the grain size of TiN tended to be uniform due to thermal activation at elevated temperatures. At the same time, Ag was separated into nanoparticles between TiN. Only a small amount of Ag was distributed on the surface. The energy dispersive spectroscopy (EDS) results showed that the stability of Ag in TiN promoted the increase in conductivity. For these two substrates, the TiN-Ag coating on SS316L exhibits better corrosion resistance and conductivity. In contrast, the aluminum was more prone to oxide film, resulting in poor adhesion and corrosion resistance of the coating. Wang et al. [29] attempted to prepare a new type of N/Ta co-doped TiO_2_ coating (TOTaN) on titanium by a sol-gel method. The results showed that the TOTaN coating enhanced the conductivity of TiO_2_ through lattice distortion caused by Ta doping and the reduction of Ti^4+^ to Ti^3+^. The ICR was about 7.1 mΩ·cm^2^. Meanwhile, the corrosion behavior of the PEMFC cathode was simulated by being immersed in 0.1 M HCl solution for 240 h. The results illustrated that TOTaN coating showed good corrosion resistance and stability under simulated cathode environmental conditions.

Therefore, the surface modification of BPs was achieved by doping other elements into the metal nitride coatings. In our previous research [30], the TiAlN coating was prepared by doping Al element in TiN. The results show that the corrosion resistance and electrical conductivity of TiAlN coatings are significantly enhanced. In general, there is little research on TiMoN coatings, so the TiMoN coatings deposited by changing the Mo target current on SS316L substrate was explored. In this experiment, TiMoN coating with excellent electrical conductivity and corrosion resistance is prepared by closed field unbalanced magnetron sputtering ion plating (CFUMSIP). X-ray diffractometer (XRD), field emission scanning electron microscopy (FE-SEM) are used to characterize the phase structure, observe the microscopic surface morphology, respectively. Potentiostatic polarization, potentiodynamic polarization, electrochemical impedance spectroscopy (EIS), and high potential (+1.2 V_SCE_) polarization measurements are used to evaluate the corrosion resistance of BPs.

## 2. Experimental Methods

### 2.1. Sample Pretreatment

In this experiment, SS316L with the diameter of 30 mm and the thickness of 3 mm is selected as the base material, whose element composition and contents are shown in Table 1. All samples are grinded with 240#, 600#, 1000#, 1500#, and 2000# SiC sandpaper before deposition. Then, the surface of the samples is polished to the mirror by using oil-soluble diamond polishing paste with the particle size of 1.0 μm and 0.5 μm, respectively. Afterward, all polished samples are ultrasonically cleaned with acetone and alcohol for 15 min, respectively. Finally, the samples are packed and placed in a dust-free, low-temperature environment.

### 2.2. Preparation of TiMoN Coatings

The deposition procedure is completed by CFUMSIP (Teer-UDP-650 coating system, London, UK). One high-purity molybdenum target (≥99.99%) and two high-purity titanium targets (≥99.99%) are used as sputtering sources. Firstly, the chamber is maintained at the base pressure of 1.5 × 10^−5^ Torr, and then the constant flow rate of Ar (≥99.99%) is launched into the chamber. Secondly, the target current changed to 0.3 A and sputtered for 30 min to eliminate pollutants and oxides on the target surfaces. Thirdly, the target current of the two titanium targets increased from 0.3 A to 6 A and deposited for 30 min to obtain a single layer of Ti, which could enhance the adhesion between the substrate and the coating. Finally, TiMoN coatings were obtained by adjusting the molybdenum target from 0.3 A to 6 A, and passing through the constant flow rate of N_2_ (≥99.99%) for 90 min. Three coatings with different Mo currents are denoted as TiMoN-2A, TiMoN-4A, and TiMoN-6 A, respectively. The complete deposition parameters are illustrated in Table 2. The deposition process diagram is shown in Figure 1.

### 2.3. Surface Characterization

To characterize the coating, X-ray diffractometer (XRD, D/max-Ultimal IV, Co., Ltd., Japan) (equipped with monochromator for copper K at 30 kV and 40 mA, scanning range 20–80°, scanning rate 20° min^−1^) was used to detect the phase structure of the TiMoN coatings and Jade6 software (Beijing, China) was used to analyze the crystal lattice constants of phases of TiMoN coatings. Field emission scanning electron microscope (FE-SEM, ƩIGMA, Carl Zeiss, UK) was used to observe the microstructure of TiMoN coatings at 20K× times. In addition, texture coefficient (TC) is usually used to express the degree of preferred orientation. The formula [31] is as follows:(1)$$\mathrm{TC}\left(\mathrm{hkl}\right)=\frac{I_{m}\left(\mathrm{hkl}\right)}{I_{o}\left(\mathrm{hkl}\right)}{\left\{\frac{1}{n}\sum_{1}^{n}\frac{I_{m}\left(\mathrm{hkl}\right)}{I_{o}\left(\mathrm{hkl}\right)}\right\}}^{-1}$$
where I_m_ (hkl), I_o_ (hkl) and n are the relative reflection intensities of the (hkl) plane of the coating and the standard powder sample, the total number of diffraction peaks, respectively.

### 2.4. Electrochemical Measurement

In this experiment, CHI660C electrochemical workstation (Chenhua, Shanghai, China) is employed for the electrochemical test, and the three-electrode system with the working electrode (coatings), reference electrode (saturated calomel electrode, SCE), and the platinum electrode is selected. The electrochemical test is conducted in the solution (bubbling air, 70 ± 2 °C 0.5 M H_2_SO_4_ + 2 ppm HF) under simulated PEMFC cathode environment. The coatings are stabilized at open-circuit potential (OCP) for 3600 s at the beginning. Then, the −0.6 V_SCE_ potential is supplied to each coating for 120 s before the electrochemical test to eliminate the oxide film on the surface of the coatings. In the voltage range of −0.6 V_SCE_ to 1.0 V_SCE_, the potentiodynamic polarization is tested at the scanning speed of 1 mV/s. The arguments obtained from the potentiodynamic polarization are corrosion potential (E_corr_), corrosion current density (I_corr_), anode tafel slope (β_a_), and cathode tafel slope (β_c_), respectively.

The polarization resistance (R_p_) is figured by the Stern–Geary [32] formula:(2)$$R_{P}=\frac{\beta_{a}{\;\times \;\beta }_{c}}{2{.303\;\times \;i}_{\mathrm{corr}}{\;\times \;(\beta }_{a}{+\beta }_{c})}$$
where β_a_ and β_c_ are the anode and cathode tafel slopes of the polarization curve, respectively.

P_i_ depicts the protection efficiency of the coatings, which is calculated from the measured corrosion current density. Here is the calculation formula [33]:(3)$$P_{i}\left(\%\right)=100\;\times (1\;–\;\frac{i_{\mathrm{corr}}}{i_{\mathrm{corr}}^{0}})$$

The long-term potentiostatic polarization test of 21,600 s is performed under simulated PEMFC cathode environment (applied +0.6 V_SCE_ potential, 0.5 M H_2_SO_4_ + 2 ppm HF, 70 ± 2 °C, bubbling air) to evaluate the stability of the BPs. Generally, PEMFC will generate high potential during the start-up/shut-down process, which accelerates the aging of the bipolar plate and shortens its life. Therefore, it is essential to investigate the corrosion resistance of BPs at high potential. In this research, the corrosion resistance test is conducted in +1.0 V_SCE_ and +1.2 V_SCE_ for 3600 s, which proves the durability of BPs. Electrochemical impedance spectroscopy measured frequency in the range of 100 kHz to 10 mHz with the amplitude of 5 mV. Zview2 software was used to fit the obtained impedance data.

### 2.5. Interfacial Contact Resistance Measurements

High durability and interfacial charge transfer capability are two important factors in evaluating the performance of PEMFC bipolar plates. The ICR between BPs and GDL is commonly used. In this experiment, the ICR between the coatings and the carbon paper is measured by the method of Kumer [34].

Under the pressure of 1.4 MPa, the ICR between the coatings and carbon paper could be calculated by the following formula:(4)$$R_{1}{=R}_{\mathrm{sample}}{+2R}_{\mathrm{cp}/\mathrm{coating}\;\;}{+2R}_{\mathrm{cp}}{+2R}_{\mathrm{cp}/\mathrm{Au}}{+2R}_{\mathrm{Au}}$$
(5)$$R_{2}{=R}_{\mathrm{cp}}{+2R}_{\mathrm{cp}/\mathrm{Au}}{+2R}_{\mathrm{Au}}$$
(6)$$R_{3}{=R}_{1}\;{–\;R}_{2}{=R}_{\mathrm{sample}}{+2R}_{\mathrm{cp}/\mathrm{coating}}{+R}_{\mathrm{cp}}\;\approx {\;2R}_{\mathrm{cp}/\mathrm{coating}}$$
(7)$${\mathrm{ICR}=R}_{\mathrm{cp}/\mathrm{sample}}=\frac{R_{3}}{2}\;\times \;S$$
where R_sample_, R_Au_, and R_cp_ represent the resistance of SS316L, gold-plated copper, and carbon paper, respectively. R_cp/coating_, R_cp/sample_, and R_cp/Au_ represent the contact resistance of carbon paper and coating, carbon paper and SS316L, carbon paper, and gold-plated copper plate, respectively. S is the effective contact area between carbon paper and sample.

### 2.6. Water Contact Angle Measurement

The water contact angles of SS316L and TiMoN coatings were measured by Dataphysics OCA30 Angle Measurement instrument. The static contact angles of SS316L and TiMoN coatings at three different positions were tested and averaged at 25 °C.

## 3. Result and Discussion

### 3.1. Phase Characterization 

Figure 2 shows the XRD patterns of TiMoN coatings and SS316L. As can be seen from the figure, all samples could find the austenite matrix peaks, concluding (111), (200), and (220) [35]. This is because the coating deposited by PVD is relatively thin, normally about 0.2–3μm, which will cause X-ray diffraction [36]. The spacing of crystal planes is the main factor for the shift of diffraction peaks, which may be caused by the change in the solid solution structure [37]. In Figure 2a, five diffraction peaks of Mo_5_N_6_ (110), TiN (111), Mo_2_N (111), TiN (200), and Ti_2_N (116) appeared. When a small amount of Mo atoms (TiMoN-2A) is doped, TiN (111), Mo_2_N (111), and Ti_2_N (116) phase can be discovered. The second strong peak of the γ-Mo_2_N (111) phase could be observed, and Ti_2_N (116) is detected at 2θ = 71.093°. This change indicates that the minority of Mo element doping will promote the phase transition. With the increase in Mo elements (4A and 6A), the stronger diffraction peak TiN (200) appeared. It can be observed from Figure 2b that γ-Mo_2_N is the subphase component, the weakest phase of Mo_5_N_6_ (110) appears, while the peak of Ti_2_N (116) almost disappears. It is worth noting that the diffraction peaks of TiN (111) and Mo_2_N (111) are shifted to the left, which indicates that Mo atomic doping has undergone the solid solution enhancement. The solid solution is related to the lattice parameters, and the lattice parameters of each diffraction peak are shown in Table 3. It can be seen from Table 3 that Mo_2_N (111) is the hexagonal crystal, while TiN (111) is the face-centered cubic crystal. The change of lattice constant indicates the solid solution strengthening, which proves Mo atoms enter the solid solution.

In addition, the transition from the diffraction peak of TiN (111) to the TiN (200) may be related to the preferred orientation. As can be seen from the entire XRD pattern, with the increase in Mo doping, the preferred orientation of the coating gradually changes from TiN (111) to TiN (200). This illustrates that TiMoN coating mainly grows preferentially in TiN (111) and TiN (200) planes. In face-centered cubic (FCC) structures, the preferred orientation crystal plane is related to the lowest free energy which is the sum of surface energy and strain energy [38]. As Mo atoms increase, the strain energy dominates. The preferred growth of TiMoN coatings along TiN (200) will reduce the total free energy.

According to Formula (1), the texture coefficient of each diffraction peak is calculated as shown in Table 4. Generally, the computed value of TC (hkl) greater than 1 may indicate the preferred orientation of the given (hkl) plane. Otherwise, it indicates random orientation or the lack of preferred orientation. According to the results in Table 4, it declares that TiMoN coatings mainly grow preferentially along with the TiN (200) direction, which is consistent with the conclusion of Wang et al. [39].

### 3.2. Microstructural Characterization

The microscopic and cross-sectional morphologies of several TiMoN coatings are shown in Figure 3. As illustrated in Figure 3a,c,e, when a small amount of Mo atoms (TiMoN-2A) is doped, the grains are significantly refined and only the surface profile can be seen. When Mo atoms gradually increase, the surface morphology changes, and TiMoN-4A has the roughest surface. In contrast, the surface morphology of TiMoN-6A is smoother than TiMoN-4A. This phenomenon may be attributed to the change in the grain growth model.

The cross-sectional morphology of the coatings is shown in Figure 3b,d,f. The thickness of the coating is 569.4 nm, 898.8 nm, and 1.329 μm, which can be seen from the cross-sectional morphology. It can be concluded from the thickness that Mo doping enhances the deposition efficiency of the coating. According to the calculation, the deposition rate has risen from 6.33 nm/min to 14.77 nm/min. TiMoN-2A has no obvious columnar crystal observed through the cross-section, which indicates that a small amount of Mo doping has changed the microstructure of the coating. In contrast, the columnar crystals of TiMoN-4A and TiMoN-6A coatings can be clearly observed. The difference is that the columnar crystal of TiMoN-4A is refined, while TiMoN-6A exhibits thick columnar crystals. Combined with the film thickness and structure analysis, TiMoN-2A coating is the thinnest and has no columnar crystal in the microstructure of the coating. TiMoN-6A coating is the thickest and has an obvious thick columnar crystal in the microstructure. However, the film thickness of TiMoN-4A coating is between the two coatings, and the columnar crystals are significantly refined. The doping of different contents of Mo atoms will cause the corrosion resistance of the coating to change, which will be verified in subsequent tests.

### 3.3. Electrochemical Corrosion Evaluation

#### 3.3.1. Potential Polarization Readings

Figure 4 depicts the potential polarization curves of SS316L and TiMoN coatings under simulated PEMFC cathode environment. The results are shown in Table 5. According to the consequences of the potentiodynamic polarization, the I_corr_ of TiMoN coatings is two to three orders of magnitude lower than SS316L and the lowest I_corr_ is 9.33 × 10^−8^ A·cm^−2^. Compared to SS316L, the corrosion potential of coatings moves to the right, which indicates that the passivation film of TiMoN coatings has been formed on the surface. With the increase in Mo content, the corrosion current density decreases first and then increases. When the Mo target current is 4A, TiMoN-4A coating is lower than other coatings, illustrating its corrosion rate in the solution is the lowest and has excellent corrosion resistance. Meanwhile, TiMoN-4A coating still maintains the lowest corrosion current density (3.57 × 10^−6^ A·cm^−2^) at 0.6 V_SCE_ potential, which is close to the target of the DOE 2020 targets. In addition, compared with other samples, the cathode slope of TiMoN-4A coating is the lowest after 0.6 V_SCE_, illustrating that TiMoN coatings have excellent high-potential corrosion resistance.

The R_p_ is crucial to judging corrosion by the potentiodynamic polarization test. Generally speaking, the greater the polarization resistance, the better the corrosion resistance [32]. The R_p_ calculated by using Formula (1) is shown in Table 5. The R_p_ of TiMoN-4A coating is 4 orders of magnitude higher than SS316L. Meanwhile, according to Formula (2), the calculation results indicate the P_i_ of the coating is higher than 95%, which significantly enhances the corrosion resistance of SS316L under simulated PEMFC cathode environment.

#### 3.3.2. Potentiostatic Polarization Readings

The electrode potential on the cathode side of PEMFCs is +0.6 V_SCE_ when the power is stable [36]. Under simulated cathode PEMFC environment (applied +0.6 V_SCE_ potential, 0.5 M H_2_SO_4_ + 2 ppm HF, 70 ± 2 °C, bubbling air), the potentiostatic polarizati on curves of SS316L and TiMoN coatings are demonstrated in Figure 5. The I_corr_ of SS316L and TiMoN coatings remain stable after dropping rapidly in the initial stage of the potentiostatic polarization due to the generation of the passivation film. Compared with the coatings, the order of the corrosion current density between them is: TiMoN-4A (5.22 × 10^−7^ A·cm^−2^) < TiMoN-6A (1.59 × 10^−6^ A·cm^−2^) < TiMoN-2A (1.97 × 10^−6^ A·cm^−2^) < SS316L (2.77 × 10^−6^ A·cm^−2^). This indicates that as the Mo element is doped, the I_corr_ of the coating increases first and then drops. The I_corr_ of TiMoN-4A coating is about 2 times the magnitude lower than SS316L, which meets the DOE 2020 targets. It means that the TiMoN-4A coating can maintain chemical stability for the long-term under simulated PEMFC cathode environment. As can be seen from Figure 5, compared with other coatings, the I_corr_ of TiMoN-6A coating in the cathode environment fluctuates, but it is still maintained at about 1.6 × 10^−6^ A·cm^−2^. It proves that it still maintains high chemical stability in the harsh cathode environment and can better resist the corrosion of the cathode environment.

#### 3.3.3. High Potential Polarization Readings

Under ideal cycling start-up/shut-down switch operation conditions, occasional load fluctuations and high frequent idling conditions, and other output power fluctuations, battery potential changes may occur within the range of +0.8∼+1.6 V_SCE_ [40,41]. The polarization curves of the SS316L and TiMoN coatings at high potential are shown in Figure 6 to assess the corrosion resistance of the coatings during the PEMFC start-up/shut-down process. As illustrated in Figure 6a, TiMoN coatings are stable after the I_corr_ declines dramatically in the period. At the high potential of 1.0 V_SCE_, TiMoN-4A coating has the lowest I_corr_, while TiMoN-2A and TiMoN-6A have similar I_corr_. According to the enlarged image, the I_corr_ of TiMoN-6A coating is slightly lower than TiMoN-2A coating. Furthermore, under the high potential of 1.0 V_SCE_, SS316L needs about 1500 s to be stable, which indicates that the formation rate of the passivation film is slow. Then, the formation and dissolution of the passivation film reach equilibrium, and the SS316L remains stable. The order of the I_corr_ of the coatings is: TiMoN-4A (3.32 × 10^−6^ A·cm^−2^) < TiMoN-6A (8.74 × 10^−5^ A·cm^−2^) < TiMoN-2A (9.07 × 10^−5^ A·cm^−2^) < SS316L (1.45 × 10^−4^ A·cm^−2^).

For Figure 6b, when the 1.2 V_SCE_ is applied, the I_corr_ of the SS316L decreases rapidly in a short period and the inflection point appears at about 70 s. The slope of the curves increases significantly, indicating that the passivation film is beginning to degrade. Additionally, TiMoN-4A and TiMoN-6A coatings show short-term stability at about 600 s and 240 s, respectively. This could be due to the production rate of the passivation film being faster than the corrosion rate of the solution in a short period. Then, TiMoN-4A and TiMoN-6A coatings experience the rapid decline of about 300 s, and finally stabilized. TiMoN-2A coating tends to be stable after declining rapidly, which may be related to the rapid formation rate of the passivation film. According to the enlarged picture, the order of the I_corr_ of the coatings is: TiMoN-4A (1.85 × 10^−5^ A·cm^−2^) < TiMoN-6A (2.15 × 10^−5^ A·cm^−2^) < TiMoN-2A (1.09 × 10^−4^ A·cm^−2^) < SS316L (8.84 × 10^−3^ A·cm^−2^).

Figure 7 depicts the surface morphology after high-potential polarization. Before corrosion, the surface morphology of the coatings is more uniform and compact compared with SS316L. After 1 h of high-potential polarization, the large area of corrosion occurs on SS316L, demonstrating that SS316L was poorly corrosive at high potential. Under high-potential corrosion, the surface morphology of the coatings is pitted in different areas. As shown in Figure 8, the pitting corrosion on the surface of TiMoN-4A coating is the lightest, while TiMoN-2A coating is the most serious. It proves that TiMoN-4A has excellent corrosion resistance under high-potential corrosion, while TiMoN-2A is the worst. Combined with the above analysis, doping with a certain content of Mo atoms will enhance the corrosion resistance of coatings at high potential, which is corresponding to the previous consequences.

#### 3.3.4. Electrochemical Impedance Spectroscopy (EIS) Measurements

Figure 8 displays the Nyquist and Bode plots of coatings under simulated PEMFC cathode environment to explain the corrosion process. Generally speaking, the high-frequency region corresponds to the electrochemical behavior of coating, and the low-frequency region corresponds to the electrochemical response of the substrate/coating interface in the Nyquist plot [42,43,44]. Generally, the wider the diameter of the semi-circular arc in the Nyquist curve, the higher the charge transfer resistance (R_ct_), indicating that the rate of metal corrosion in the acid solution is lower [36]. As shown in Figure 8a, the semicircular arc diameter of TiMoN coatings is larger than SS316L, indicating that the corrosion rate of the TiMoN coatings in the acid solution is relatively low. The order of the corrosion rate in the cathode environment is TiMoN-4A < TiMoN-6A < TiMoN-2A < SS316L.

Figure 8b,c depicts the bode diagrams of all coatings. In general, in the Bode phase diagram, the wider the frequency region, the more phase content, which indicates the better passive behavior. The high-frequency area, middle-frequency area, and low-frequency area represent the contact characteristics between the solution interface and the coating, the corrosion inside the coating, and the contact characteristics between the coating and the substrate, respectively [16]. Generally, it is considered that the greater the low-frequency region |Z|, and the wider the peak width of the medium frequency region, the better the corrosion resistance [45]. In addition, the phase angle of TiMoN coatings is close to 90° in the middle- and low-frequency region, indicating that the coating has two-time constants. However, the phase angle of SS316L is close to 90° in a very narrow frequency region, indicating that the SS316L has fewer time constants than TiMoN coatings [46].

In the low-frequency region, the |Z| value of TiMoN coatings is larger than SS316L, which proves TiMoN coatings have excellent corrosion resistance. The impedance |Z| sequence at the fixed frequency of 10 mHz is: TiMoN-4A > TiMoN-6A > TiMoN-2A > SS316L. This proves that TiMoN-4A coating has the best corrosion resistance under the simulated PEMFC cathode environment.

To analyze the impedance results quantitatively, the equivalent circuit model is established by fitting the impedance results through Zview2 software. The fitting results are shown in Table 6. The equivalent circuit model in Figure 8a (1) is used to analyze the corrosion behavior of SS316L. This model has been widely used by researchers before [47]. The corrosion resistance of two-time constant coatings is commonly described by graph (2) in Figure 8a [16]. Among them, where R_s_, R_coat_, R_ct_, CPE_dl_, and CPE_coat,_ respectively, represent the solution resistance, coating resistance, charge transfer resistance, double-layer constant origin, and coating constant origin. According to the fitting results in Table 6, the R_ct_ sequence of SS316L and TiMoN coatings is: TiMoN-4A (41,572 Ω·cm^2^) > TiMoN-6A (26,748 Ω·cm^2^) > TiMoN-2A (15,596 Ω·cm^2^) > SS316L (900.9 Ω·cm^2^). Generally, the larger the R_ct_ values, the lower electrochemical reactian rate. This indicates TiMoN-4A coating could better prevent the corrosive ions from the substrate, which has the best protection characteristics under simulated PEMFC cathode environment.

Through the above analysis, TiMoN coatings have good protective effect than SS316L, and can effectively prevent the corrosion ions from the substrate, which achieves good anti-corrosion effect. Furthermore, with the increase in Mo doping, the corrosion resistance first increases and then decreases. TiMoN-4A coating exhibits the best anti-corrosion effect among all samples, which is consistent with the previous analysis.

### 3.4. ICR Measurements

The resistance generated between BPs and GDL in PEMFC is called interfacial contact resistance (ICR), which is important data for estimating the conductivity [27,30]. Figure 9 depicts the ICR of SS316L and TiMoN coatings at pre-polarization, 0.6 V_SCE_, and 1.2 V_SCE_, respectively. As shown in Figure 9, the ICR of SS316L before corrosion is 120.52 mΩ·cm^2^. With the increase in Mo content, the conductivity of TiMoN coatings has been significantly enhanced. Compared with SS316L, it has dropped by about 6–12 times. With the increase in Mo content, the ICR of TiMoN coatings decreases first and then increases. The order of the ICR of TiMoN coatings is TiMoN-4A (9.19 mΩ·cm^2^) < TiMoN-6A (16.14 mΩ·cm^2^) < TiMoN-2A (19.79 mΩ·cm^2^). Among them, TiMoN-4A coating meets DOE 2020 targets. After +0.6 V_SCE_ polarization, the ICR of SS316L increase to 215.24 mΩ·cm^2^, while the ICR of TiMoN coatings increases by about 11 (±1) mΩ·cm^2^. This indicates that under the protection of TiMoN coatings, BPs still have good conductivity. After +1.2 V_SCE_ polarization, the ICR of TiMoN coatings is further enhanced. TiMoN-4A coating has the lowest ICR of 39.23 mΩ·cm^2^. According to the above analysis, when the Mo target current is 4 A, the coating has the best effect on the conductivity of BPs.

ICR is mainly affected by surface state and film composition [48]. The effective contact area between the carbon paper and the coating surface is the main factor affecting the surface condition. The larger effective the contact area, the more conducive to electronic transmission, and the smaller the ICR [16]. Combined with the micro-morphological analysis in Figure 3, the rough surface of TiMoN-4A coating increases the effective contact area, which promotes the active movement of electrons and the formation of conductive channels [49]. The preferred orientation of TiMoN-4A coating changes from TiN (111) to TiN (200), which has excellent conductivity according to the results of the XRD analysis. Therefore, the preparation of TiMoN coating is beneficial to improve the conductivity of SS316L BPs. When the Mo target current is 4A, the coating has the lowest ICR.

### 3.5. Water Contact Angle Measurements

The water contact angle of SS316L and TiMoN coatings are shown in Figure 10. When the static contact angle is greater than 90°, which represents hydrophobic. The hydrophobic film can reduce the residence time of the corrosive solution on the surface of the bipolar plate and destroy the corrosive environment formed on the surface [43]. The order of the water contact angle of TiMoN coatings is TiMoN-4A > TiMoN-6A > TiMoN-2A > SS316L. The values are 107.0°, 105.6°, 104.8°, 69.7°, respectively. This indicates TiMoN coatings have good hydrophobicity and can extend the service life of BPs.

## 4. Conclusions

In this research, TiMoN coatings were deposited on SS316L substrate by CFUMSIP technology. By changing the Mo target current, three kinds of TiMoN coatings with different currents were prepared, named as TiMoN TiMoN-2A, TiMoN-4A and TiMoN-6A, respectively. Here are the main conclusions:(1)From the XRD analysis, When the Mo target current is 2A, the phases of TiMoN coating are mainly TiN (111) and Mo2N (111), and the weak phase Ti2N (116). With the increase in Mo target current (4A and 6A), the phase structure of TiMoN coating changes. The phase structure of TiMoN coating is mainly TiN (200), and the phase of Mo5N6 is generated. In addition, according to the texture coefficient and lattice lattice parameters, it is proved that the change of Mo element causes the solution strengthening of TiMoN coating, and Mo atom enters TiN solid solution;(2)Combined with FE-SEM surface and cross-sectional micro-morphology analysis, When the Mo target current is 2A, the microstructure of TiMoN coating is uniform but relatively thin. With the current of Mo target increases, the surface morphology of TiMoN coating becomes rough, and the thickness increases. The deposition rate increases from 6.33 nm/min to 14.77 nm/min by calculation;(3)Potentiodynamic polarization and potentiostatic polarization tests show that the Icorr of TiMoN coatings is better than SS316L substrate. At the same time, the Icorr of TiMoN coatings meets DOE 2020 targets (≤1 × 10^−6^ A·cm^−2^). Among them, TiMoN-4A coating has the lowest corrosion current density, which is 9.33 × 10^−8^ A·cm^−2^ and 5.22 × 10^−7^ A·cm^−2^, respectively. The high-potential polarization test shows that TiMoN-4A coating has the lowest Icorr at high potential (+1.0 VSCE, +1.2 VSCE). Meanwhile, the surface morphology of TiMoN-4A coating has no obvious change, indicating that TiMoN-4A coating has good corrosion resistance under high-potential corrosion;(4)The results of EIS show that the semi-arc resistance of TiMoN coating increases obviously, indicating that the corrosion resistance is enhanced. Meanwhile, the phase angle of TiMoN coating is close to 90° in the range of middle- and low-frequency region, indicating that the TiMoN coatings have two-time constants;(5)The ICR results show that the ICR of TiMoN coatings is significantly lower than SS316L. Meanwhile, the ICR of TiMoN-4A coating is the lowest of 9.19 mΩ·cm^2^, which meets the DOE 2020 targets (≤10 mΩ·cm^2^);(6)The water contact angle test results show that the water contact angle of TiMoN coatings is significantly higher than 90°, which indicates that TiMoN coatings have good hydrophobicity. Among them, the TiMoN-4A coating has the largest water contact angle and the best hydrophobicity.

## Figures and Tables

**Figure 1 materials-15-03766-f001:**
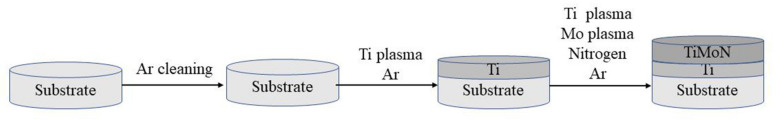
Deposition process of TiMoN coating.

**Figure 2 materials-15-03766-f002:**
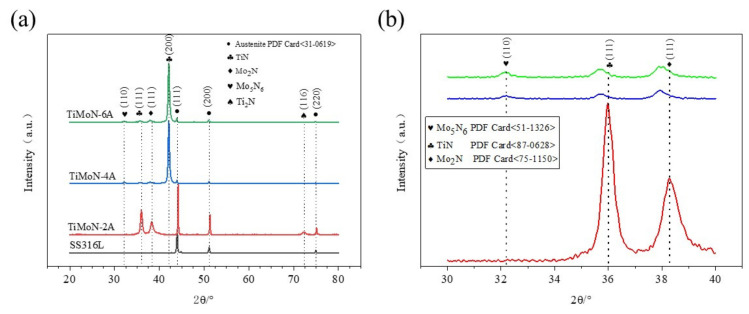
XRD pattern of SS316L and TiMoN coatings: (**a**) full spectrum, (**b**) partial spectrum.

**Figure 3 materials-15-03766-f003:**
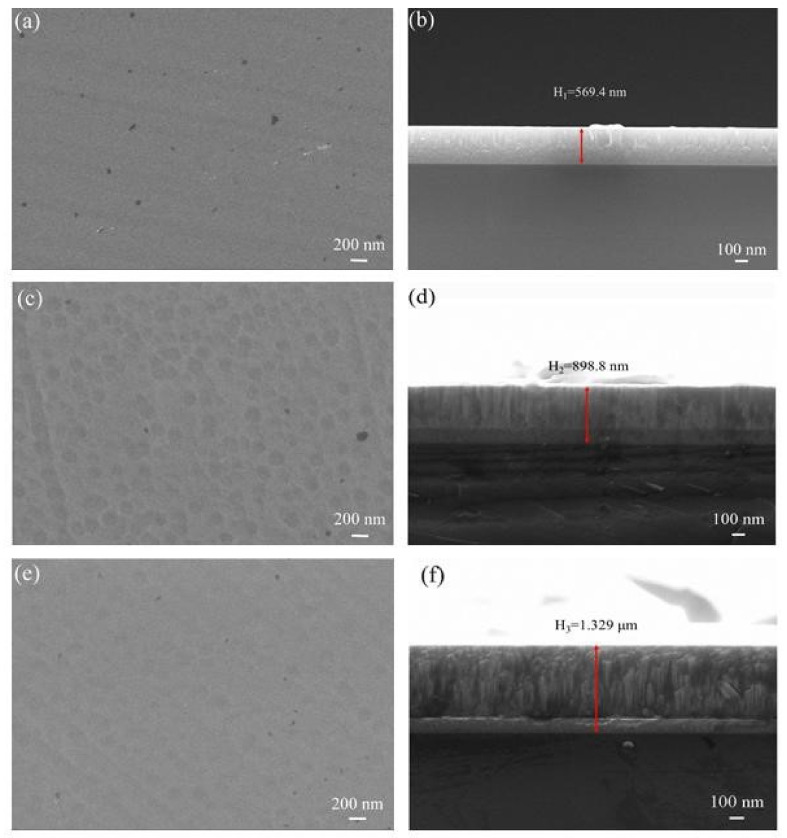
The FE-SEM electron microscope image of the microscopic and cross-section of TiMoN coatings: (**a**,**b**) TiMoN-2A, (**c**,**d**) TiMoN-4A, (**e**,**f**) TiMoN-6A.

**Figure 4 materials-15-03766-f004:**
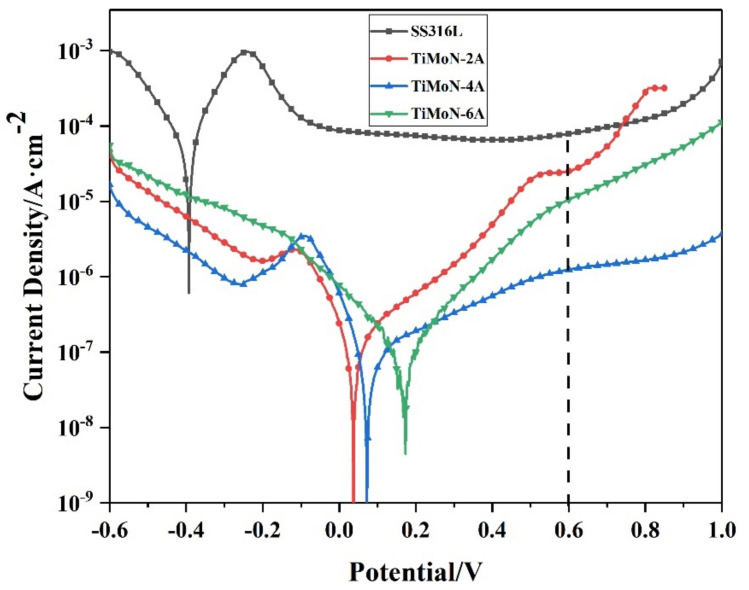
Potentiodynamic polarization curves of SS316L and TiMoN coatings under simulated PEMFC cathode environment.

**Figure 5 materials-15-03766-f005:**
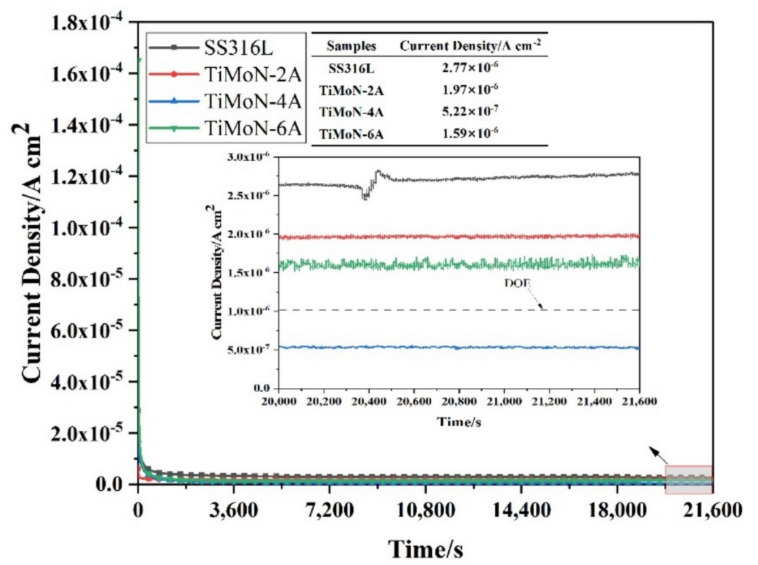
Potentiostatic polarization curves of SS316L and TiMoN coatings under simulated PEMFC cathode environment.

**Figure 6 materials-15-03766-f006:**
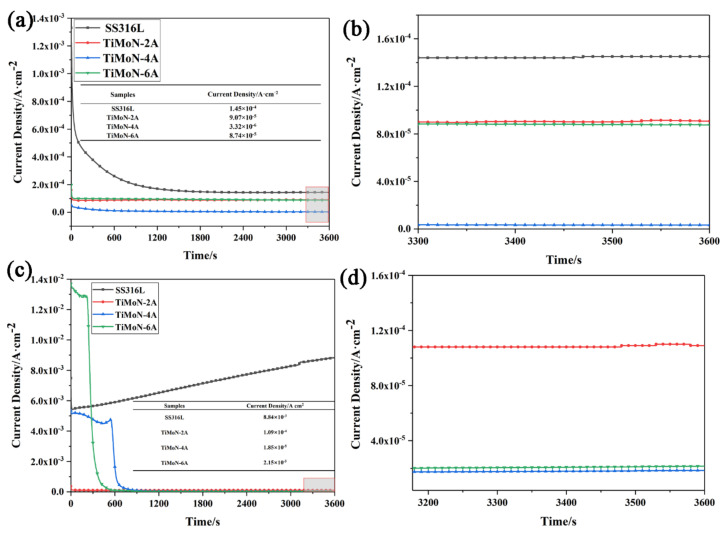
The high-potential polarization curves of SS316L and TiMoN coatings under a simulated PEMFC cathode environment. (**a**) 1.0 V_SCE_, (**b**) 1.0 V_SCE_ enlarge image, (**c**) 1.2 V_SCE_, (**d**) 1.2 V_SCE_ enlarge image.

**Figure 7 materials-15-03766-f007:**
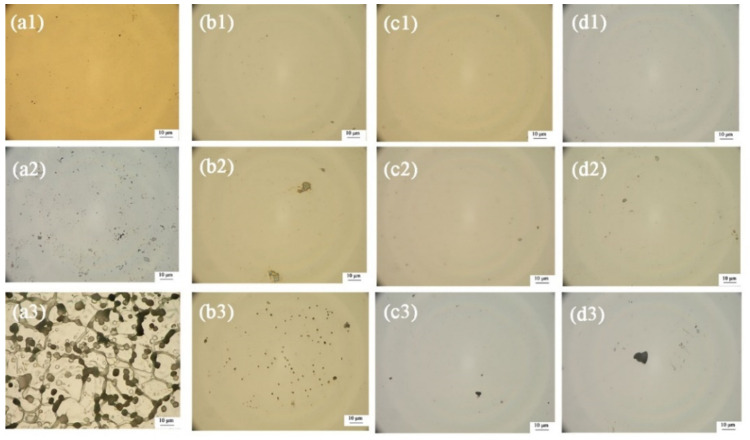
Surface morphology of SS316L and TiMoN coatings before and after high-potential polarization (**a1**–**d1**) pre-polarization, (**a2**–**d2**) 1.0 V_SCE_, (**a3**–**d3**) 1.2V_SCE_; (**a1**–**a3**) SS316L, (**b1**–**b3**) TiMoN-2A, (**c1**–**c3**) TiMoN-4A, (**d1**–**d3**) TiMoN-6A.

**Figure 8 materials-15-03766-f008:**
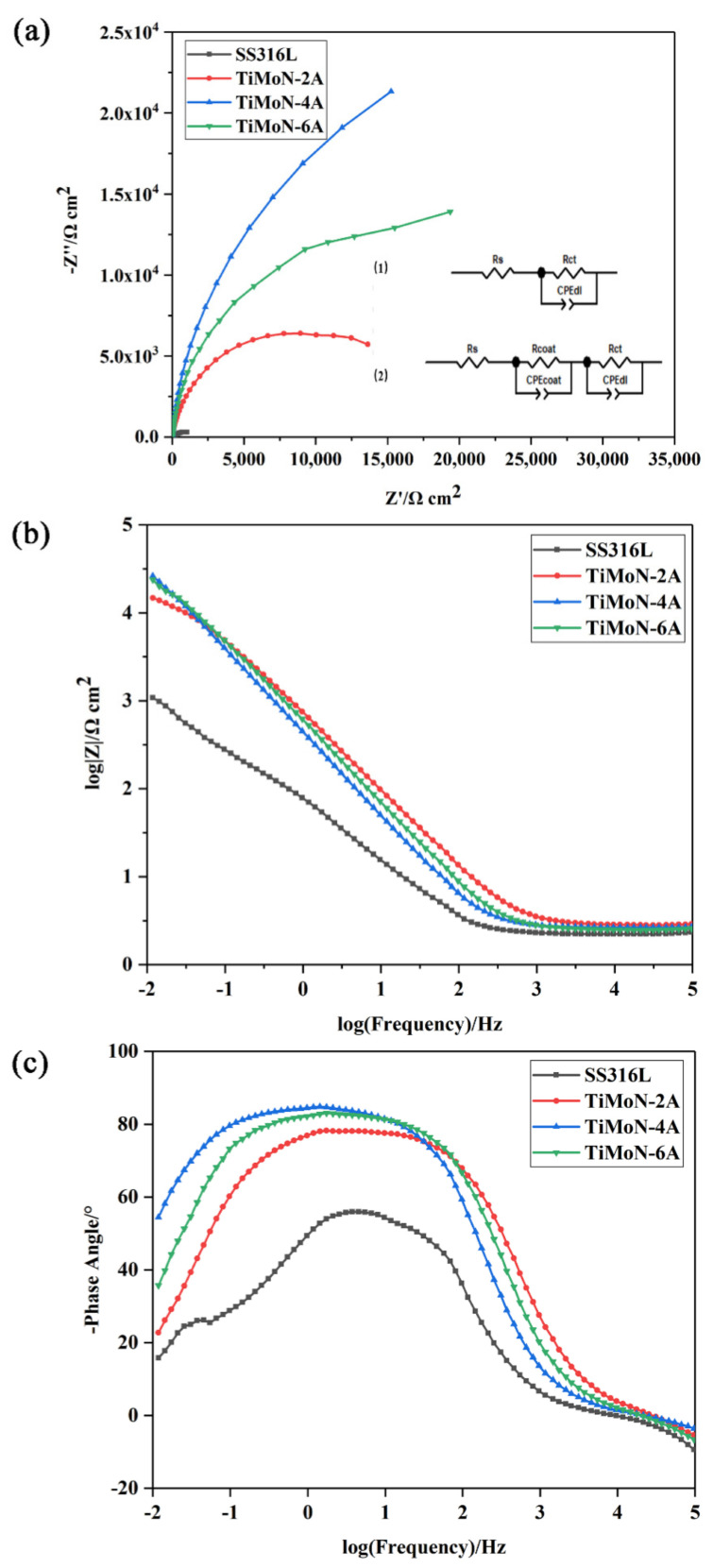
EIS curves of SS316L and TiMoN coatings: (**a**) Nyquist plots, (**b**) |Z| plots, (**c**) phase angle plots.

**Figure 9 materials-15-03766-f009:**
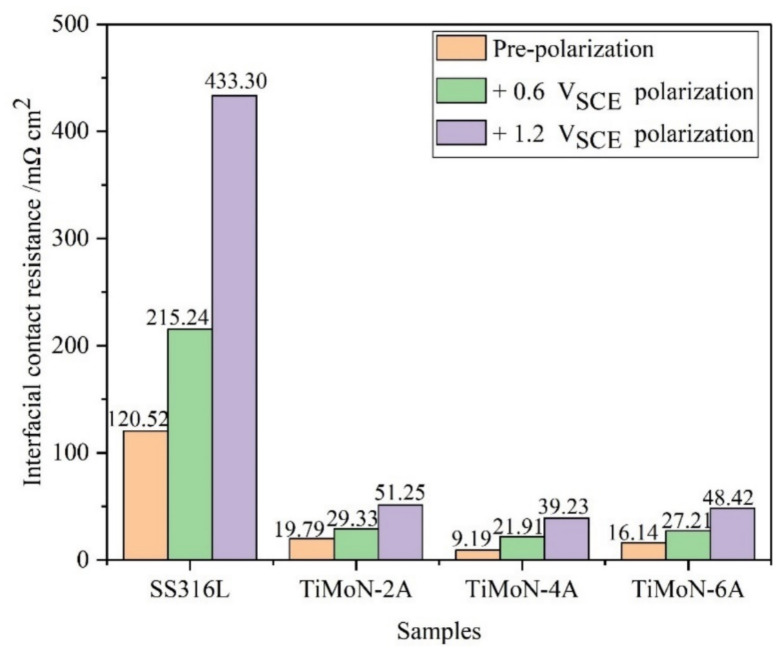
ICR measurement results of SS316L and TiMoN coatings before corrosion, 0.6 V_SCE_ and 1.2 V_SCE_ at high potential under simulated PEMFC cathode environment.

**Figure 10 materials-15-03766-f010:**
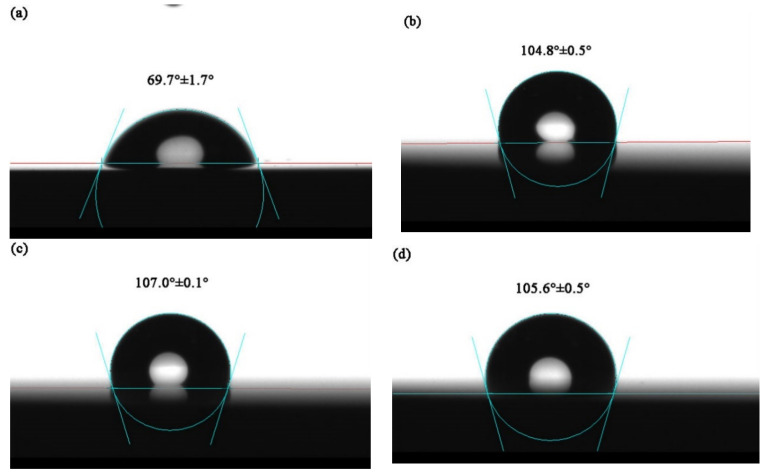
Water contact angle of SS316L and TiMoN coatings: (**a**) SS316L, (**b**) TiMoN-2A, (**c**) TiMoN-4A, (**d**) TiMoN-6A.

**Table 1 materials-15-03766-t001:** Element composition and content of SS316L (wt.%).

Cr	Ni	Mo	Mn	Si	P	S	C	Fe
20.0–21.0	6.0–7.0	1.5	1.5	1.0	0.04	0.03	0.03	Balance

**Table 2 materials-15-03766-t002:** TiMoN coating deposition process parameters.

Deposition Process	Ar Gas Flow Rate/Sccm	N2 Gas Flow Rate/Sccm	Ti Target Current/A	Mo Target Current/A	Bias Voltage/V	Deposition Time/s
1. Surface cleaning	40	0	0.3	0.3	−500	1800
2. Ti layer deposited	40	0	6	0.3	−80	1800
3. TiMoN coating	40	30	6	2, 4, 6	−80	5400

**Table 3 materials-15-03766-t003:** Lattice constants of the main diffraction peaks.

Samples	Mo_5_N_6_			TiN			Ti_2_N			Mo_2_N		
	Lattice Constant/nm											
	a	b	c	a	b	c	a	b	c	a	b	c
TiMoN-2A	/	/	/	0.2956	0.2956	0.4756	0.4188	0.4188	0.8048	0.4210	0.4210	0.8060
TiMoN-4A	0.4893	0.4893	1.106	0.4244	0.4244	0.4244	/	/	/	0.4140	0.4140	0.8805
TiMoN-6A	0.7605	0.7605	1.062	0.4320	0.4320	0.4320	/	/	/	0.4210	0.4210	0.8060

**Table 4 materials-15-03766-t004:** Texture coefficients of different planes of TiMoN coatings.

Samples	Crystal Plane				
		Mo_5_N_6_ (111)	TiN (111)	Mo_2_N (111)	TiN (200)
TiMoN-2A	TC (hkl)	/	0.735	0.524	/
TiMoN-4A		0.171	0.054	0.130	1.356
TiMoN-6A		0.153	0.041	0.118	1.287

**Table 5 materials-15-03766-t005:** Fitting consequences of potentiodynamic polarization of SS316L and TiMoN coatings.

Samples	E_corr_/V	I_corr_/A·cm^−2^	Β_a_/V dec^−1^	β_c_/V dec^−1^	R_p_/ohm	I_0.6V_/A·cm^−2^	P_i_/%
SS316L	−0.397	4.48 × 10^−5^	0.126	0.084	4.48 × 10^2^	7.81 × 10^−5^	/
TiMoN-2A	0.029	1.42 × 10^−7^	0.096	0.267	2.16 × 10^6^	2.51 × 10^−6^	99.68
TiMoN-4A	0.081	9.33 × 10^−8^	0.094	0.416	3.57 × 10^6^	1.25 × 10^−6^	99.79
TiMoN-6A	0.169	1.34 × 10^−7^	0.210	0.213	3.43 × 10^6^	1.06 × 10^−6^	99.70

**Table 6 materials-15-03766-t006:** EIS fitting consequences of SS316L and coatings.

Samples	R_s_/Ω·cm^2^	CPE_coat_-T/cm^−2^S^−n^Ω	CPE_coat_-P/cm^−2^S^−n^Ω	R_coat_/Ω cm	CPE_dl_-T/cm^−2^S^−n^Ω	CPE_dl_-P/cm^−2^S^−n^Ω	R_ct_/Ω·cm^2^
SS316L	2.10	/	/	/	3.44 × 10^−3^	0.732	900.9
TiMoN-2A	2.83	1.82 × 10^−3^	0.873	714.7	3.09 × 10^−4^	0.888	15,596
TiMoN-4A	2.65	2.46 × 10^−3^	0.818	1021	4.45 × 10^−4^	0.997	41,572
TiMoN-6A	2.51	1.76 × 10^−3^	0.853	1107	3.50 × 10^−4^	0.959	26,748

## Data Availability

The data is available on reasonable request from the corresponding author.

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
