# Peer review of "Study on Corrosion Resistance and Conductivity of TiMoN Coatings with Different Mo Contents under Simulated PEMFC Cathode Environment"

_materials, 2022, doi:10.3390/ma15113766_

Round 1
Reviewer 1 Report
The paper entitled “Study on corrosion resistance and conductivity of TiMoN coatings with different Mo contents under simulated PEMFC cathode environment” focuses on the deposition and examination of TiMoN coatings with different Mo contents on SS316L substrate by using Closed Field Unbalanced Magnetron Sputtering Ion Plating (CFUMSIP) technology. The analysis of the obtained films includes X-ray diffraction (XRD) examinations, SEM with EDX analysis, electrochemical tests (potentiodynamic, potentiostatic, high potential polarization, and EIS), the interfacial contact resistance measurements, and water contact measurements.
Although the focus of the paper is interesting, I would like to recommend the rejection of the paper for publication because of some main reasons.
- In the abstract, the abbreviations of the samples are unclear to the readers. They should be explained.
- The equation for calculating texture coefficient (TC) (7) should be moved to section 2.
- The water contact measurements have not been presented in section 2.
- The TC should be calculated for each sample and each phase. The results presented did not indicate any tendency and have no meaningful explanation.
- It would be better for the authors in Tables 3 and 4 to compare the lattice parameters and texture coefficients, respectively, of each phase in the three different samples. In table 3, the reference values could also be given.
- The change in crystalline size can be calculated and presented depending on the increase of the Mo target current. The grain size could be determinative of the electrochemical performance of the coatings.
- The statement that in the FCC system, (200) plane exhibits lower “surface energy and strain energy than (111) plane” is not true. For this statement, the authors refer to ref.[36] where such a statement can not be found. In ref. 36 it is said that “In face-centered cubic (FCC) structures, the (200) and (111) planes have the lowest surface energy and strain energy, respectively. When the strain energy is dominant, the preferred growth of film along (111) plane will reduce the total free energy”. Actually, (111) orientation of TiN films exhibits the lowest strain energy [Hai-juan MEI, et al/Trans. Nonferrous Met. Soc. China 28(2018) 1368−1376]
- The data presented in Table 5 seems statistically unreliable. The level of statistical significance (standard deviations of values) of the measurements should be presented.
- A clear explanation of the reason for the lower Mo content in TiMoN-6A should be given.
- The authors said that “compared with other samples, the slope of TiMoN-4A coating is the lowest after 0.6 VSCE, illustrating that the coating has excellent high potential corrosion resistance”. It is not clear which slope they meant – the cathodic or the anodic?
- The influence of the structure and texture of the coatings, as well as their thickness, should also be commented on when considering the electrochemical results.
- The statement “the micro-morphological analysis in Fig.3, the rough surface of TiMoN-4A coating increases the effective contact area” should be confirmed with additional examinations. Roughness measurements by AFM for example could be further held.
- The conclusions can re-formulated. They sound more like a summary of the results rather than a conclusion.
- Some technical comments:
- In some places, the English style and grammar have to be improved. Some examples of inadequate style: “ In face-centered cubic (FCC) structures, the (200) plane has the lowest surface energy and strain energy than the (111) plane”; “In the low-frequency region, the Ç€ZÇ€ of SS316L is smaller than TiMoN coatings, which proves TiMoN coatings have excellent corrosion resistance.” …and some more….
– The sentence “According to the fitting results in Table 6, the 366 sequence of TiMoN coatings and SS316L is: TiMoN-4A (41572 Ω·cm2)>TiMoN-6A (26748 367Ω·cm2)>TiMoN-2A (15596 Ω·cm2)>SS316L (900.9 Ω·cm2).” has unclear meaning.
Author Response
Author Response to Reviewers
June 6, 2022
Editorial Office
Materials
Manuscript Number: materials-1708900
Manuscript title: Study on corrosion resistance and conductivity of TiMoN coatings with different Mo contents under simulated PEMFC cathode environment
Dear Editors and Reviewers:
Thank you for your letter and for the reviewers' comments concerning our manuscript entitled " Study on corrosion resistance and conductivity of TiMoN coatings with different Mo contents under simulated PEMFC cathode environment " (Manuscript Number: materials-1708900). Those comments are all valuable and very helpful for revising and improving our paper, as well as the important guiding significance to our researches. We have studied comments carefully and have made correction which we hope meet with approval. The main corrections in the paper and the responds to reviewers' comments are as flowing:
Responds to the reviewers' comments:
Reviewer #1:
Response to comment:
- In the abstract, the abbreviations of the samples are unclear to the readers. They should be explained.
- The equation for calculating texture coefficient (TC) (7) should be moved to section 2.
- The water contact measurements have not been presented in section 2.
- The TC should be calculated for each sample and each phase. The results presented did not indicate any tendency and have no meaningful explanation.
- It would be better for the authors in Tables 3 and 4 to compare the lattice parameters and texture coefficients, respectively, of each phase in the three different samples. In table 3, the reference values could also be given.
- The change in crystalline size can be calculated and presented depending on the increase of the Mo target current. The grain size could be determinative of the electrochemical performance of the coatings.
- The statement that in the FCC system, (200) plane exhibits lower “surface energy and strain energy than (111) plane” is not true. For this statement, the authors refer to ref [36] where such a statement cannot be found. In ref. 36 it is said that “In face-centered cubic (FCC) structures, the (200) and (111) planes have the lowest surface energy and strain energy, respectively. When the strain energy is dominant, the preferred growth of film along (111) plane will reduce the total free energy”. Actually, (111) orientation of TiN films exhibits the lowest strain energy [Hai-juan MEI, et al/Trans. Nonferrous Met. Soc. China 28(2018) 1368−1376]
- The data presented in Table 5 seems statistically unreliable. The level of statistical significance (standard deviations of values) of the measurements should be presented.
- A clear explanation of the reason for the lower Mo content in TiMoN-6A should be given.
- The authors said that “compared with other samples, the slope of TiMoN-4A coating is the lowest after 0.6 VSCE, illustrating that the coating has excellent high potential corrosion resistance”. It is not clear which slope they meant – the cathodic or the anodic?
- The influence of the structure and texture of the coatings, as well as their thickness, should also be commented on when considering the electrochemical results.
- The statement “the micro-morphological analysis in Fig.3, the rough surface of TiMoN-4A coating increases the effective contact area” should be confirmed with additional examinations. Roughness measurements by AFM for example could be further held.
- The conclusions can re-formulated. They sound more like a summary of the results rather than a conclusion.
- Some technical comments:
- In some places, the English style and grammar have to be improved. Some examples of inadequate style: “In face-centered cubic (FCC) structures, the (200) plane has the lowest surface energy and strain energy than the (111) plane”; “In the low-frequency region, the Ç€ZÇ€ of SS316L is smaller than TiMoN coatings, which proves TiMoN coatings have excellent corrosion resistance.” …and some more….
The sentence “According to the fitting results in Table 6, the Rct sequence of TiMoN coatings and SS316L is: TiMoN-4A (41572 Ω·cm2)>TiMoN-6A (26748 367Ω·cm2)>TiMoN-2A (15596 Ω·cm2)>SS316L (900.9 Ω·cm2).” has unclear meaning.
Response: Thanks for your comment. We have modified the paper according to the suggestions of reviewers. The modification details are as follows:
- For recommendations 1-5, I think it is important in the opinions of reviewers, which have modified according to the opinions of reviewers.
- For recommendations 6, thank you very much for your advice. Grain size is an important factor in determining electrochemical performance, but I think it can also be proved to be related to electrochemical performance in other ways, such as texture coefficient.
- For recommendations 7,in ref. 36(After the revision, it is ref 38) it is said that “In face-centered cubic (FCC) structures, the (200) and (111) planes have the lowest surface energy and strain energy, respectively. When the strain energy is dominant, the preferred growth of film along (111) plane will reduce the total free energy”. In reference 38, the author made the analysis according to the XRD pattern. I think it is not quite right to use his conclusion to justify my conclusion, because his XRD pattern is different from mine. What I introduced was only his viewpoints on strain energy and surface energy, and then carried out practical analysis according to my own XRD patterns. In my XRD pattern, according to formula 1, TiN (200) shows the largest preferred orientation. The results in Table 3 and Table 4 have proved the solid solution enhancement occurs with the Mo element doped.it is concluded that the strain energy dominates and the preferred growth of TiMoN coatings along TiN (200) will reduce the total free energy.
- For recommendations 8-9, The EDS results in Table 5 have no standard deviation because the occurrence of Mo atomic content percentage proves that Mo atoms have entered the solid solution. Therefore, the reason why the Mo atom content in TiMoN-6A is low is that the solid solution reaches saturation, so the Mo content decrease slightly.
- For recommendations 10, The vehicle generates the high potential of 0.84-1.6V during the start-up/shut-down process(ref.40-41), while the TiMoN coatings can reach 1.2V after 0.6VSCE. The slope of TiMoN-4A is the lowest after 0.6 VSCE under the simulated cathode environment of PEMFC in this paper, so it shows good high potential corrosion resistance.
- For recommendations 11, The structure and thickness of TiMoN coatings have been analyzed by XRD and FE-SEM. Meanwhile, the deposited TiMoN coatings still have excellent electrochemical performance when it is damaged by potentiostatic polarization, potentiodynamic polarization and high-potential polarization. Therefore, the thickness and structure of TiMoN coatings are demonstrated in combination with the electrochemical performance.
- For recommendations 12, I am so sorry that we are unable to conduct AFM testing due to COVID-19. In addition, I have found relevant paper (ref.48), which points out that the increase of roughness is conducive to the increase of effective contact area. I think the ref.48 proves that roughness can increase the effective contact area in this paper.
- For recommendations 13-14, Thank you for your suggestions. We have modified corrections according to your suggestions.
Finally, we appreciate the work of editors and reviewers, and hope that the revision will meet your requirements.
Sincerely Yours,
Rui Cao

Reviewer 2 Report
The author present a well structured study of the TiMoN thin films coating onto SS316L samples. The coating was varied by increase the Mo content during the CFUMSIP . The resulting samples were characterized by XRD, SEM, EDS and contact angle. The electrochemical corrosion was studied by potentiostatic polarization and EIS. The corrosion resistance was improved due to the TiMoN coating, and a better corrosion protection was obtained for samples prepared at 4A-Mo target current.
I have some comments about the manuscript:
section 3.2: the authors describe the grain and roughness of the samples. But, it is hard to observe the grains from the SEM images. On the other hand, the roughness can be obtained/discussed from topographic images only, for example AFM images.
Figures 6 and 7: Because the current density is decreasing in a logaritmic fashion i recomend graph it in a logaritmic scale.
Figure 8: The scale bar is not legible.
Figure 9: check the units
Table 7: check the units
Section 3.5: How many measurements of the contact angle for each sample were performed? Please include the error bar.
Section 4: you must rewrite the conclusions. The actual version is a simple collection of results.
Author Response
Author Response to Reviewers
June 2, 2022
Editorial Office
Materials
Manuscript Number: materials-1708900
Manuscript title: Study on corrosion resistance and conductivity of TiMoN coatings with different Mo contents under simulated PEMFC cathode environment
Dear Editors and Reviewers:
Thank you for your letter and for the reviewers' comments concerning our manuscript entitled " Study on corrosion resistance and conductivity of TiMoN coatings with different Mo contents under simulated PEMFC cathode environment " (Manuscript Number: materials-1708900). Those comments are all valuable and very helpful for revising and improving our paper, as well as the important guiding significance to our researches. We have studied comments carefully and have made correction which we hope meet with approval. The main corrections in the paper and the responds to reviewers' comments are as flowing:
Responds to the reviewers' comments:
Reviewer #2:
Response to comment:
section 3.2: the authors describe the grain and roughness of the samples. But, it is hard to observe the grains from the SEM images. On the other hand, the roughness can be obtained/discussed from topographic images only, for example AFM images.
Figures 6 and 7: Because the current density is decreasing in a logaritmic fashion i recomend graph it in a logaritmic scale.
Figure 8: The scale bar is not legible.
Figure 9: check the units
Table 7: check the units
Section 3.5: How many measurements of the contact angle for each sample were performed? Please include the error bar.
Section 4: you must rewrite the conclusions. The actual version is a simple collection of results.
Response: Thanks for your comment. We have modified the paper according to the suggestions of reviewers. The modification details are as follows:
- For recommendations 1: We only consider the surface morphology to be rough, because it is different from the flat and dense surface we studied before. In addition, I am so sorry that we are unable to conduct AFM testing due to COVID-19. We are tring to prove the roughness of the surface in as many ways as possible in future papers.
- For recommendations 2: In fig.6 and fig.7, we have made corresponding modifications according to the suggestions of reviewers. We read a lot of papers related to PEMFC bipolar plates to observe and learn their painting mode. Thank you for your valuable suggestions. We will refer to your comments for continuous improvement and strive to break through in the paper drawing.
- For recommendations 3-5: Thanks for the valuable suggestions of reviewers. We have made modifications as required.
- For recommendations 6: Normally, the water contact angles were tested three or four times at different locations of each TiMoN coating. The water contact angle obtained is averaged. We have modified the water contact angles according to the suggestions of reviewers.
- For recommendations 7: We have modified the conclusion according to the suggestions of reviewers
Finally, we appreciate the work of editors and reviewers, and hope that the revision will meet your requirements.
Sincerely Yours,
Rui Cao

Reviewer 3 Report
The authors studied the influences of Mo on the PVD coatings formation and properties on top of SS316L substrate. This research is in the scope of the journal of Materials; however, it does not have very high novelty, although the authors claim their results new/different compared to the literature. I have some suggestions to make the paper improved, and if the editor tends to consider this paper for publication, a major revision is needed.
1) All abbreviations must first be explained in the introduction (despite the abstract). For example, BPs, ICRs, etc.
2) There is no justification for the choice of coating composition/system by authors, its potential advantages and disadvantages against to other known coatings which descibed in the Introduction. Why the system TiMoN was chosen by the authors as a research system ? And why they decided to vary the content of molybdenum in the coating? The problematics and aim of the work are not clearly formulated and need to be improved.
3)Table 1 - the ACTUAL chemical composition of the substrate on which the coating samples were obtained should be given.
4) I recommend to the authors provide all parameters for XRD measurements (scanning step, scan geometry) and SEM investigations (parameters missing)
5) The quality of the XRD analysis needs to be improved for such thin coatings. The XRD should be made in GIXRD mode at low speed with small step (0.01deg or less) and the results and discussion should be added to the manuscript.
6) How did the authors perform a full-profile analysis of XRD patterns and determine the lattice parameters? The authors should add to the methodological section these details of the lattice refinement procedure as well as measurements errors of the lattice parameters.
7) I strongly recommend to the authors to calculate the parameters of the microstructure of the obtained coatings (crystallite size, residual microstrains, dislocation density) and to provide and discuss them in the paper.
8) Scale bars in fig. 3 are very small and difficult to read. I recommend to the authors to increase them to perhaps 250 or maybe 500 nm.
9) The authors presented composition of the coatings basing on EDX analysis, including nitrogen. It is well known that EDX is not proper method for quantitative analysis of nitrogen (and other light elements with atomic number up to 11) concentration and it this paper the mistake presented frequently in other papers should not be duplicated. Here you can find the information about limitations of quantitative EDX analysis: https://myscope.training/legacy/analysis/eds/quantitative/
Please, use another analytical method for nitrogen analysis.
10) The authors should revise and supplement Table 6. First, I strongly recommend that the authors not use the XE±Y notation. I recommend to the authors use well-known prefixes (numerical prefixes - uA, mkA & etc) or a power of 10. Secondly, in Table 6, the authors do not give measurement errors and it is not possible to assess the importance of, for example, the decrease in icorr c 1.42*10^(-7) A*cm(-2) to 9*10^(-8)A*cm(-2) for TiMoN-2A and TiMoN-4A coatings, respectively. Similarly, for other characteristics. The measurement errors for all data should be given in the Table 6.
11) The results given in sections 3.2.2 and 3.3.3 of the paper without measurement errors or repeatability (for several samples) are unreasonable. It is necessary to give measurement errors or show the stability of the current characteristics for several samples of each type.
12) The values of the corrosion current densities in the inserts in Figs. 7 are extremely small and difficult to read. They should be made larger or removed from the Figs.
13)The scale bars on the Fig 8 should be increased.
14) Table 7 - The errors for the individual parameters of the equivalent electrical circuits (such as CPE and R) should be provided.
15) The statement “Among them, TiMoN-4A coating meets DOE 2020 targets” is unreasoned without indicating the measurement error for ICR equal to 9.19 mΩ cm2 against required <10mΩ cm2
16) The error bars on the Fig. should be given.
17) I strongly recommend to the authors to give more discussion and compare the obtained results for their coatings with the known results of similar and/or competitive coatings; show the advantages and disadvantages of their samples, analyze and discuss possible ways for improving the performance of the proposed TiMoN coatings.
Author Response
Author Response to Reviewers
June 6, 2022
Editorial Office
Materials
Manuscript Number: materials-1708900
Manuscript title: Study on corrosion resistance and conductivity of TiMoN coatings with different Mo contents under simulated PEMFC cathode environment
Dear Editors and Reviewers:
Thank you for your letter and for the reviewers' comments concerning our manuscript entitled " Study on corrosion resistance and conductivity of TiMoN coatings with different Mo contents under simulated PEMFC cathode environment " (Manuscript Number: materials-1708900). Those comments are all valuable and very helpful for revising and improving our paper, as well as the important guiding significance to our researches. We have studied comments carefully and have made correction which we hope meet with approval. The main corrections in the paper and the responds to reviewers' comments are as flowing:
Responds to the reviewers' comments:
Reviewer #3:
Response to comment:
1) All abbreviations must first be explained in the introduction (despite the abstract). For example, BPs, ICRs, etc.
2) There is no justification for the choice of coating composition/system by authors, its potential advantages and disadvantages against to other known coatings which descibed in the Introduction. Why the system TiMoN was chosen by the authors as a research system ? And why they decided to vary the content of molybdenum in the coating? The problematics and aim of the work are not clearly formulated and need to be improved.
3)Table 1 - the ACTUAL chemical composition of the substrate on which the coating samples were obtained should be given.
4) I recommend to the authors provide all parameters for XRD measurements (scanning step, scan geometry) and SEM investigations (parameters missing)
5) The quality of the XRD analysis needs to be improved for such thin coatings. The XRD should be made in GIXRD mode at low speed with small step (0.01deg or less) and the results and discussion should be added to the manuscript.
6) How did the authors perform a full-profile analysis of XRD patterns and determine the lattice parameters? The authors should add to the methodological section these details of the lattice refinement procedure as well as measurements errors of the lattice parameters.
7) I strongly recommend to the authors to calculate the parameters of the microstructure of the obtained coatings (crystallite size, residual microstrains, dislocation density) and to provide and discuss them in the paper.
8) Scale bars in fig. 3 are very small and difficult to read. I recommend to the authors to increase them to perhaps 250 or maybe 500 nm.
9) The authors presented composition of the coatings basing on EDX analysis, including nitrogen. It is well known that EDX is not proper method for quantitative analysis of nitrogen (and other light elements with atomic number up to 11) concentration and it this paper the mistake presented frequently in other papers should not be duplicated. Here you can find the information about limitations of quantitative EDX analysis: https://myscope.training/legacy/analysis/eds/quantitative/
Please, use another analytical method for nitrogen analysis.
10) The authors should revise and supplement Table 6. First, I strongly recommend that the authors not use the XE±Y notation. I recommend to the authors use well-known prefixes (numerical prefixes - uA, mkA & etc) or a power of 10. Secondly, in Table 6, the authors do not give measurement errors and it is not possible to assess the importance of, for example, the decrease in icorr c 1.42*10^(-7) A*cm(-2) to 9*10^(-8)A*cm(-2) for TiMoN-2A and TiMoN-4A coatings, respectively. Similarly, for other characteristics. The measurement errors for all data should be given in the Table 6.
11) The results given in sections 3.2.2 and 3.3.3 of the paper without measurement errors or repeatability (for several samples) are unreasonable. It is necessary to give measurement errors or show the stability of the current characteristics for several samples of each type.
12) The values of the corrosion current densities in the inserts in Figs. 7 are extremely small and difficult to read. They should be made larger or removed from the Figs.
13)The scale bars on the Fig 8 should be increased.
14) Table 7 - The errors for the individual parameters of the equivalent electrical circuits (such as CPE and R) should be provided.
15) The statement “Among them, TiMoN-4A coating meets DOE 2020 targets” is unreasoned without indicating the measurement error for ICR equal to 9.19 mΩ cm2 against required <10mΩ cm2
16) The error bars on the Fig. should be given.
17) I strongly recommend to the authors to give more discussion and compare the obtained results for their coatings with the known results of similar and/or competitive coatings; show the advantages and disadvantages of their samples, analyze and discuss possible ways for improving the performance of the proposed TiMoN coatings.
Response: Thanks for your comment. We have modified the paper according to the suggestions of reviewers. The modification details are as follows:
- For recommendations 1-2: I think it is important in the opinions of reviewers, which have modified according to the opinions of reviewers.
- For recommendations 3: Since the commercial SS316L rod was selected, the actual chemical composition is given in Table 1.
- For recommendations 4: Thank you for your suggestions. I have provided the specific parameters of XRD and SEM tests in section 2.3(Surface characterization)
- For recommendations 5: The XRD diffractometer that we use is carried out under the fixed range and cannot be changed. In addition, we focus on analyzing the phase structure of TiMoN coatings rather than the specific parameters of the test.
- For recommendations 6: Jade6 software was used to analyze the crystal lattice constants of phases of TiMoN coatings and we have added this to Section 2.3(Surface characterization).
- For recommendations 7: Thanks for the suggestions of reviewers. I am so sorry that we are unable to conduct microstructure parameters (microcrystalline size, residual microstress and dislocation density) due to COVID-19. We will try to add them in future papers.
- For recommendations 8: According to our previous papers published in other journals, FE-SEM was used to observe the microstructure of the TiMoN coatings g at 20Kx times. According to the scale conversion, the surface is 200nm and the cross section is 100nm.
- For recommendations 9: Thank you for your proposal. According to the element composition of TiMoN coating, we tried to detect Ti, Mo and N elements and obtained the atomic percentage instead of using EDS to analyze nitrogen as the reviewer said. the percentage of atoms was used to determine whether the content of Mo element changes, so as to prove whether Mo atom enters the solid solution. We looked for previous our papers published in other journals, such as Corrosion Science (Investigation of high potential corrosion protection with titanium carbonitride coating on 316L stainless steel bipolar plates), International Hydrogen Energy (Formation of a protective TiN layer by liquid phase plasma electrolytic nitridation on Ti–6Al–4V bipolar plates for PEMFC) etc. We use the similar methods to solve the data obtained by EDS.
- For recommendations 10: Thank you for your suggestions. We have modified Table 6 according to the requirements of reviewers. In addition, the current density of PEMFC is defined as 1.0×10-6 A·cm-2 in DOE 2020 targets, which the measured data only need to be compared DOE 2020 targets.
- For recommendations 11:In response to the questions raised by reviewers in sections 3.3.2 and 3.3.3, measurement errors are very important in the experimental process. We refer to the drawing methods of articles published by other authors in related fields for drawing. However, considering the particularity of electrochemical testing process, it is impossible to test the same sample in the same experimental state at the same time. Moreover, the potentiodynamic polarization test destroys the coating surface. the measurement error in electrochemical testing basically does not fluctuate much. There is almost no significant fluctuation of measurement error in electrochemical test. Therefore, the experimental method of our default test is feasible.
- For recommendations 12-13:In order to avoid the problem that reviewers could not see the pictures clearly, we used high-definition pictures and the data in the pictures were explained with words. As for fig.8, We have made modifications as required.
- For recommendations14: Table 7 shows the fitting results obtained by the equivalent circuit model and Zview2 software, rather than artificially calculated data, which has been controlled to the reasonable range of errors in Zview2 software. The fitting error of the control is between 10% and 15%.
- For recommendations15: According to DOE 2020 targets, ICR should meet the requirements of less than 10 mΩ·cm2. All the ICR measurements were tested more than three times of experimental results, so the influence of experimental errors could be ignored.
- For recommendations16-17: Normally, the water contact angles were tested three or four times at different locations of each TiMoN coating. The water contact angle obtained is averaged. We have modified the water contact angles according to the suggestions of reviewers. We will try our best to combine pictures with bar charts in future papers; In addition, we have modified the conclusion according to the opinions of reviewers.
Finally, we appreciate the work of editors and reviewers, and hope that the revision will meet your requirements.
Sincerely Yours,
Rui Cao

Round 2
Reviewer 1 Report
The authors have addressed some of the review’s recommendations and the manuscript has been improved. Just one more comment, please improve the sentence "In addition, compared with other samples, the slope of TiMoN-4A coating is the lowest after 0.6 VSCE, illustrating that TiMoN coatings have excellent high potential corrosion resistance.". The words "the slope" should be "the slopes", "the cathodic slope" or "the anodic slope".
Author Response
Author Response to Reviewers
June 8, 2022
Editorial Office
Materials
Manuscript Number: materials-1708900
Manuscript title: Study on corrosion resistance and conductivity of TiMoN coatings with different Mo contents under simulated PEMFC cathode environment
Dear Editors and Reviewers:
Thank you for your letter and for the reviewers' comments concerning our manuscript entitled " Study on corrosion resistance and conductivity of TiMoN coatings with different Mo contents under simulated PEMFC cathode environment " (Manuscript Number: materials-1708900). Those comments are all valuable and very helpful for revising and improving our paper, as well as the important guiding significance to our researches. We have studied comments carefully and have made correction which we hope meet with approval. The main corrections in the paper and the responds to reviewers' comments are as flowing:
Responds to the reviewers' comments:
Reviewer #2:
Response to comment:
The authors have addressed some of the review’s recommendations and the manuscript has been improved. Just one more comment, please improve the sentence "In addition, compared with other samples, the slope of TiMoN-4A coating is the lowest after 0.6 VSCE, illustrating that TiMoN coatings have excellent high potential corrosion resistance.". The words "the slope" should be "the slopes", "the cathodic slope" or "the anodic slope".
Response: Thanks for your comment. We have modified the paper according to the suggestions of reviewers. The modification details are as follows:
For recommendations 1: We have modified for the opinion of reviewers. Thank you very much for the reviewers.
Reviewer 3 Report
In my opinion, the authors have given satisfactory responses to the comments and made some improvements in the paper.
However, I am not satisfied with the answers of the authors regarding the impossibility of calculating the parameters of the microstructure and residual microstrains in the obtained samples from the already existing XRD patterns.
I hope that the authors will pay more attention to this in their future researches and works .
In addition, I believe that authors should produce a group of samples of 3 or 5 or 7 or 10 pieces in each deposition mode, test them and take into account the statistical spread of characteristics to improve the reliability and reproducibility of research results.
I did not see this in this work and, in my opinion, this significantly reduces its value and importance.
I also consider the lack of a detailed discussion and comparison of the results obtained by the authors with well-known works and competitive solutions as a disadvantage of the work, which did not allow the authors to show the significant advantages of their technology.
Author Response
Author Response to Reviewers
June 8, 2022
Editorial Office
Materials
Manuscript Number: materials-1708900
Manuscript title: Study on corrosion resistance and conductivity of TiMoN coatings with different Mo contents under simulated PEMFC cathode environment
Dear Editors and Reviewers:
Thank you for your letter and for the reviewers' comments concerning our manuscript entitled " Study on corrosion resistance and conductivity of TiMoN coatings with different Mo contents under simulated PEMFC cathode environment " (Manuscript Number: materials-1708900). Those comments are all valuable and very helpful for revising and improving our paper, as well as the important guiding significance to our researches. We have studied comments carefully and have made correction which we hope meet with approval. The main corrections in the paper and the responds to reviewers' comments are as flowing:
Responds to the reviewers' comments:
Reviewer #2:
Comments and Suggestions for Authors
In my opinion, the authors have given satisfactory responses to the comments and made some improvements in the paper.
However, I am not satisfied with the answers of the authors regarding the impossibility of calculating the parameters of the microstructure and residual microstrains in the obtained samples from the already existing XRD patterns.
I hope that the authors will pay more attention to this in their future researches and works .
In addition, I believe that authors should produce a group of samples of 3 or 5 or 7 or 10 pieces in each deposition mode, test them and take into account the statistical spread of characteristics to improve the reliability and reproducibility of research results.
I did not see this in this work and, in my opinion, this significantly reduces its value and importance.
I also consider the lack of a detailed discussion and comparison of the results obtained by the authors with well-known works and competitive solutions as a disadvantage of the work, which did not allow the authors to show the significant advantages of their technology.
Response: Thanks for your comment. We have modified the paper according to the suggestions of reviewers. The suggestions details are as follows:
- It's hard to be convincing thatcalculating the parameters of the microstructure and residual microstrains in the obtained samples from the already existing XRD patterns. However,we cannot prove that the characteristics of microstructure and XRD patterns in TiMoN coatings by AFM and EBSD tests due to the COVID-19. The suggestions of reviewers are very pertinent, and we will pay attention to these problems in future papers.
- As for the reviewer's suggestion, I think it is of great statistical and scientific significance. However, it is difficult to repeat experiments for statistical and scientific studies for the same sample (for example, potentiodynamic polarization and potentiostatic polarization will damage the sample surface). In the following papers, we strive to achieve the law of experimental repeatability the minimum experimental error.
- The suggestions of reviewers in the conclusion part are very pertinent. The imperfect conclusion is due to the limitation of the author's writing level. I will try my best to improve my writing level and summarizing ability in the future, so as to make my articles more scientific and complete.
Finally, we appreciate the work of editors and reviewers, and hope that the revision will meet your requirements.
Sincerely Yours,
Rui Cao
